# Roots, Riots, and Radical Change—A Road Less Travelled for Ecological Economics

**Elke Pirgmaier** *[ID] and **Julia K. Steinberger**

School of Earth and Environment, University of Leeds, Leeds LS2 9JT, UK; j.k.steinberger@leeds.ac.uk
* Correspondence: e.pirgmaier@leeds.ac.uk

**Abstract:** In this paper, we put forward a new research agenda for ecological economics, based on three realisations. We then show how these can be connected through research and used to generate insights with the potential for application in broader, systemic change. The first realisation is that the core ambition of ecological economics, that of addressing the scale of human environmental resource use and associated impacts, often remains an aspirational goal, rather than being applied within research. In understanding intertwined environmental and social challenges, systemic approaches (including system dynamics) should be revitalised to address the full scope of what is possible or desirable. The second realisation is that the focus on biophysical and economic quantification and methods has been at the expense of a comprehensive social understanding of environmental impacts and barriers to change—including the role of power, social class, geographical location, historical change, and achieving human well-being. For instance, by fetishising growth as the core problem, attention is diverted away from underlying social drivers—monetary gains as profits, rent, or interest fuelled by capitalist competition and, ultimately, unequal power relations. The third realisation is that ecological economics situates itself with respect to mainstream (neoclassical) economics, but simultaneously adopts some of its mandate and blind spots, even in its more progressive camps. Pragmatic attempts to adopt mainstream concepts and tools often comfort, rather than challenge, the reproduction of the very power relations that stand in the way of sustainability transitions. We consider these three realisations as impediments for developing ecological economics as an emancipatory critical research paradigm and political project. We will not focus on or detail the failings of ecological economics, but state what we believe they are and reformulate them as research priorities. By describing and bringing these three elements together, we are able to outline an ambitious research agenda for ecological economics, one capable of catalysing real social change.

**Keywords:** capital; value; well-being; planetary boundaries; systems perspectives; degrowth; Marxian Political Economy; systems of provision; neoclassical economics

---

"Two roads diverged in a wood, and I—
I took the one less traveled by,
And that has made all the difference."
-Robert Frost

## 1. Introduction

With few years left to avert looming environmental and societal breakdown, we may ask what point there is to write a piece about a future research agenda of ecological economics. Do we need more research? Probably not. At the same time, in our capacity as researchers, most of us working at universities and unwilling to hide our heads in the sand, we can ask what we as ecological economists can contribute to tackling the biggest societal challenge we are facing: How do we live relatively peacefully as a global community, in a world of accelerating ecological overshoot? The core task

is to support building resilient communities and societies, able to prioritise the well-being of their members without durably damaging the biosphere, and to prevent the escalation of social conflicts and, ultimately, wars.

Ecological economics provides several promising foundations to address this task, in terms of vision, ambition, and approach. We are a problem-oriented community, with political aspirations towards goals of social justice and equality. We prioritize human needs and well-being [1,2], and therefore study distribution conflicts [3], power and vested interests [4], institutional change [5,6], and environmental values and ethics [7]. We offer a methodological tool-kit fit for studying social ecological complexities in realistic and transformative ways. This requires systemic thinking (understanding interrelations, emergence, and co-evolution) [8]; interdisciplinarity (adopting different perspectives and mixed-method approaches) [9]; transdisciplinarity (engaging in deliberative and participatory approaches) [10]; biophysical and social assessments (e.g., input–output modelling, material flow analyses, sustainability indicators) [11]; value pluralism and incommensurability (multi-criteria decisions, value articulating institutions) [6,12]; and a post-normal understanding of science (reflecting strong uncertainty, complexity, reflexivity) [13]. These methodological elements demarcate ecological economics from environmental economics. Moreover, our field hosts spaces for advancing radical transformations, such as degrowth ideas [14].

At the same time, we find that the ambitions and approaches that ecological economists set out for themselves often do not translate all the way through to actual research, remaining aspirational rather than applied. For instance, we claim to be the field that studies the interrelations between natural and economic systems, but in practice we rarely study monetary, social, and biophysical flows in parallel. Or, we claim to embrace systems thinking as a fundamental methodology, but then stop doing so when it comes to core economic categories, such as value, prices, or profits where we either adopt neoclassical reasoning or give up on economics altogether. Good intentions are not good enough. Research can either support or impede radical changes, and by falling back into static and piecemeal approaches, research can be *harmful.* It is harmful, when it justifies and supports policies and actions that reproduce the status quo, which is unsustainable and unjust, instead of fundamentally challenging it. The role of science in legitimizing a social order is crucial, and this holds especially for a problem-oriented discipline with close links to economics [15].

This paper responds to the call for papers for this Special Issue to contribute elements of an ambitious research agenda for ecological economics. In Section 2, we highlight three areas where ecological economics research needs improvement. First, ecological economists often fail to ground the aspirational goal of addressing the scale of human environmental impacts in applied research. Second, the strong emphasis on biophysical analysis (symptoms) and economic growth (outcomes) fails to address the underlying social root causes of ecological destruction. Third, retaining strong ties to neoclassical economics approaches is problematic, because they reinforce the status quo and delay radical action. We reach these conclusions by considering the journal *Ecological Economics* in particular, but also the wider ecological economics literature. In Section 3, we show how these realisations can be transformed into research priorities looking forward:

1.  **Face reality**. Promote the application of Marxian Political Economy as a realistic economic theory and systemic methodology for understanding capitalist dynamics in both research and teaching (textbooks, summer schools).
2.  **Stop harmful economics**. Resist the reproduction of mainstream economic narratives, distorted ideas, and pragmatic tools by exposing them as serious distractions and barriers to desirable systemic change.
3.  **Confront power**. Oppose capitalist institutions that uphold a highly destructive money-making machinery for a global elite based on structural inequality and exploitation (e.g., megabanks, fossil corporations).

4. **Prioritise what matters**. Design and support institutions whose purpose is the direct provision of human needs and dignity—access to healthy food, clean water, mobility, healthcare, and education; rather than indirect provision via growth, job-creation, and profit-making.

5. **Act**. Encourage academic collaborative action and bravery, that is, use academic spaces to pursue all of the above.

We need to fight the old (2,3), create the new (4,5), starting from the here and now (1), and do it on the fly, very fast. We see these elements as part of a progressive research and action agenda in ecological economics that have existed on fringes but have yet to become mainstream. Section 4 concludes by encouraging ecological economists to take this road less travelled.

## 2. Three Realisations about the State of the Art of Ecological Economics

### 2.1. Becoming Serious about Planetary Scale

The issue of scale is a vexing one in ecological economics. On the one hand, it is foundational—it lies at the heart of the full earth/empty earth dichotomy and is the first of Herman Daly's three principles: Biophysical flows related to the economy should remain within environmental limits. Several notable efforts have resulted from this emphasis. The measurement of the social metabolism of the economy and the macro input and output of biophysical flows related to economic activities in physical units constitutes an immense achievement—and sits squarely at the intersection of the research interests of ecological economics and industrial ecology [16–21].

Moreover, the interface between specific flows or types of resource extraction, land-use change, pollution, and environmental impacts has also gained in terms of research methods and insights [22]. The entire area of environmental limits has been revitalised under the banner of planetary boundaries [23,24], with a new historical dimension brought in through the concept of the Anthropocene [25,26]. One area where there are still considerable research gaps is the causal pathway between planetary boundaries and macro-economic biophysical resource use. In the case of climate change, it is extremely well-documented and understood, but much less so for biodiversity. Overall, the area of environmental limits, and what can be expected if we continue to transgress these is well established [27–29].

What is lacking, sadly, is on the socio-economic side of ecological economics. As a community, we are still more comfortable addressing issues of equitable distribution and efficient processes than considering scale itself. When scale is considered, it is usually through the support for resource caps as policies, as in the case of Steady-State Economics. Simply calling for such policies, without considering the fundamental barriers to their promotion, adoption, and consequences, is insufficient—it is akin to a mathematician assuming an axiom which she needs to prove.

The now well-established degrowth community comes closest to addressing macro-economic scale issues full-frontally [14,30,31]. They do this mostly implicitly, by acknowledging that the current biophysical scale of economies is well beyond environmental limits, and thus advocating for a significant decrease (degrowth) in their biophysical scale. Moreover, the degrowth community openly acknowledges and addresses the fact that such a biophysical decrease can only be accomplished through a radical decline in the scale of the monetary economy itself. The mission of degrowth economics is to highlight alternative social and economic structures and institutions [14] which prioritise societal and environmental preservation over economic imperatives. Effectively, degrowth seeks to make human societies growth-proof. However, the issue of addressing biophysical scale aside from degrowing industrial economies is not given much attention.

We argue that economic and social efforts that address the issue of biophysical scale directly should be a core focus of ecological economics, regardless of one's affinity for degrowth, steady-state or green growth. When we consider scale of biophysical flows full-frontally, several aspects become immediately clear:

1. Different biophysical flows (for instance crop products vs. sand vs. metals vs. fossil fuels) need to be considered and studied separately. Here, we argue that this can and should be done on the basis of interference in planetary biogeochemical cycles [32] and planetary boundaries [19,23,33].

2. These different biophysical flows generally correspond to different economic sectors, products, and consumption categories. We are thus no longer talking in the vague comfort of a general macro-economy, but getting into the nitty gritty of supply chains, international trade relations of extraction–manufacturing–consumption [34], and specific sectors and firms. This means that ecological economics must grapple with what a biophysical scale reduction or elimination of resource use (or considerations of transitioning to different types of resources and technologies) means for specific sectors, products, and types of consumption.

3. These specificities and areas of research cannot be addressed through neoclassical economic tools involving gradual shifts in costs and benefits, aggregate supply and demand, and so on. They demand attention to specific actors, interest groups, and their relationships. They require a new arsenal of methods and approaches, including, crucially, ones from sociology and political economy. The Systems of Provision approach of Ben Fine is relevant here, because it provides a methodological framework that embeds the study of concrete and context-specific biophysical and cultural realities in an understanding of capitalism as a whole [35].

*2.2. From Biophysical Growth to Social and Political Aspects of Change*

The second realisation is the flip side of the previous one: By framing our intertwined environmental and social predicament predominantly in biophysical terms, ecological economists remain fairly close to a neoclassical mechanistic understanding of the economic process (see Section 2.3), where the circular flow of commodities has been supplemented with some biophysical (and entropically dissipative) pipes. Whilst the biophysical approach to understanding economic processes is insightful in many respects (for instance, to understand the reliance on certain types of resources and their impacts on ecosystems, especially for dynamics and scale effects), it is insufficient for an in-depth understanding of root causes of intertwined social ecological crises. Biophysical accounts are crucial for substantiating ecological overuse, but cannot explain ecological destruction at a fundamental level. They provide evidence for what is happening (humanity entering the Anthropocene), but fail to explain *why*. A causal understanding is necessary to comprehend the magnitude of social, political, and economic changes required for a 'safe(r) environmental space' in the future, and even more so for devising viable strategies to attain those changes.

One manifestation of the neglect by ecological economics of deeper-seated social drivers is the fetishization of growth as the core problem. Why is it that ecological economists problematize growth—in biophysical and economic terms—prominently and forcefully as a key problem underpinning various kinds of ecological destructions? This problem framing might be a legacy of early general systems theory, which remains closely tied to the origins and foundations of ecological economics. At the time when systems theories were developed in the 1960s, systems were conceptualised as black boxes. Because the inner life of systems is complex and not transparent, regularities can be observed by scrutinising inputs and/or outputs. As Kenneth Boulding, the co-founder of General Systems Theory, states:

> *"A system is a big black box*
>
> *Of which we can't unlock the locks,*
>
> *And all we can find out about*
>
> *Is what goes in and what goes out."*
>
> Boulding cited in [8] p. 88

Attempts to influence unknown system behaviour are then made by varying inputs or outputs. This implies a shift away from the system itself to its surroundings. The most remarkable observable

event or 'output' of the economic system is its growth in relation to the surrounding biosphere. This core observation arose out of early systems modelling, as Donella Meadows confirms:

> *"Asked by the Club of Rome to show how major global problems—poverty and hunger, environmental destruction, resource depletion, urban deterioration, unemployment—are related and how they might be solved, Forrester made a computer model and came out with a clear leverage point: Growth."* [8] p. 1

The growth-as-core-problem framing became firmly cemented in ecological economics. It has led to much research measuring biophysical inputs, throughput, and output, and conducting analyses that pit core conflicts between the growth of 'the economy' and 'the environment'. It gave rise to ecological economic conceptions of the economic system as a growing economic subsystem black box, as typically portrayed on the first few pages of textbooks (e.g., [36]). It also led to core critiques of growth-economics and growth-economies (e.g., [37]). To this day, it shapes core debates that centre on 'managing without growth' [38], achieving 'prosperity without growth' [39], moving towards degrowth societies [31], remaining agnostic about growth [40,41], or adopting green growth strategies [42]. This leads us to posit that ecological economics might itself be caught up in a growth fetish.

Our observation is that ecological economists overemphasise growth in economic and biophysical terms at the expense of underpinning social drivers of ecological destruction. Growth is an emergent outcome of the system, not its fundamental driver. *"What is more important for understanding the behaviour of a system are its interrelation and underlying purpose or function"* [43] p. 88. Why do capitalist economies expand as part of their normal reproduction? What accumulates, when capital accumulates? These questions remain under-addressed in ecological economics. Typically, in ecological economics, the real fuel of economic growth is considered to be the extraction and use of coal, oil, gas, and natural resources as well as technological advancement. These are intermediate rather than fundamental drivers of ecological overshoot and social crises. We need to dig deeper and understand what motivates technological dynamism and the extremely wasteful and harmful use of energy and resources.

The root causes of global environmental changes are coupled social and technical phenomena, not natural ones. The social system that dominates the world is not 'the economy', i.e., a system detached from time and space; but capitalism, i.e., a historically specific class-based system of extraction, transformation, and organisation of flows of commodities for exchange in markets. Capitalism centres, at its heart, at the production of value on an extended scale, fuelled by the systemic need to accrue profits and survive under capitalist competition. A clear understanding of the capitalist core remains surprisingly vague or implicit in ecological economics.

An understanding of 'capital' offers a coherent and compelling causal explanation for the 'growing economic subsystem' that lies at the centre of ecological economists' concerns. Capital was understood long ago by Marx [44] as self-expanding value, and capital, not growth, lies at the core of the problem. Capital is defined as value in motion or self-expanding value and can thus only be understood on the basis of value theory. Value theory in turn requires an understanding of the crucial role of labour exploitation (surplus value creation), a concept that is woefully underemphasised in ecological economics [45]. An understanding of the categories value, surplus value, capital, exploitation, domination, profit, and capitalist competition is crucial, because these explain core drivers of the system, including fuel energy, land and resource exploitation, and thus environmental destruction. This understanding specifies the systemic challenge and helps justify the need for radical changes (see Section 3.2).

The omission of a realistic economic theory that identifies root causes is a serious problem in ecological economics (on the role of economic theory in ecological economics see Sections 2.3 and 3.2), because it prevents an open confrontation with capitalist institutions that stand in the way of sustainability transformations. Key considerations, in this regard, are agency and power: These are deliberately neglected in neoclassical economics and have yet to become central concerns in ecological economics. There have been key contributions from environmental justice and conflict studies [3,46],

focusing on the social conflicts at locations of extraction. Recently, Doris Fuchs and colleagues brought a much needed reminder of the power held by different actors along supply chains [4]. But ecological economics as a field has not embraced the notion of placing social relations and social conflict at the heart of our analyses of the environmental consequences of economic activities.

### 2.3. Better Economic Foundations for Better Decisions

The relationship between ecological economics and neoclassical economics has been highly contradictory, as is well-known. Whilst our field came into being out of a deep concern that standard economic applications to the environment are insufficient to effectively deal with modern environmental crises [47,48], many ecological economists adopt mainstream theory, tools, and techniques, often in the pursuit of policy and academic influence. As neoclassical ideas shape policy-debates, close ties are seen as desirable by many. But why is policy and politics dominated by mainstream economics? And why are 95% of the economics discipline dominated by one school of thought? Certainly not because they offer better economics. We cannot understand neoclassical dominance without the existing distribution of power in society. Theory and praxis go hand in hand.

By upholding a narrative and set of beliefs of capitalism as being the best of all possible worlds, and of capitalist markets as mechanisms to yield desirable social outcomes that at most require marginal fixes, mainstream economists support the reproduction of the system, rather than critically exposing and challenging it. This narrative, which has become deeply enshrined in our lives, prevents us from having an honest debate about the foundations of our societies. As such, neoclassical economics stands in the way of what ecological economists want to achieve—a transformation of society towards environmental and social justice. Rather than being a helpful and transparent tool for radical change, it is a protective belt around capital. Many heterodox economists have warned of the severe distortions and real-life consequences of mainstream economics (e.g., [49,50]), and several ecological economists have argued for the need to let go of mainstream foundations altogether [51–54]. We fully support their view.

Our main point here—and this is our third observation—is that neoclassical theory and reasoning sit much deeper in ecological economics than is often assumed. We find that even some of the most progressive thinkers in our field, who aim to redirect ecological economics along heterodox lines, often do not go far enough in linking research to key issues at stake. Much of the ecological economics agenda remains tied to *extending* the realm of neoclassical economics, rather than rejecting it as unrealistic, unscientific, and ideological and moving on to propose radical alternatives commensurate with contemporary challenges. Neoclassical 'extensions' remain prominent, for instance, when conceptualising the economic system as a whole. By characterising the economic process as a transformation of matter–energy into goods and services, ecological economists fall back into the mainstream economics conception of the economic process as essentially a barter (or simple exchange-based) economy:

> *"The economy is a complex process that converts raw materials (and energy) into useful goods and final services."* [55] p. 2

This sounds intuitive, but, for understanding the general working of the system as a whole, it is misleading. Capitalist production is driven by the exchange of money quantities, not (physical) commodities, and the reproduction of money quantities on an extended scale [44]. Commodities (and their underpinning matter–energy stocks and flows) *enable* this process of monetary expansion, rather than motivate it. By studying the 'real' (biophysically-based) economy, ecological economists implicitly adopt and extend the neoclassical economic conception of a 'real' economy (in the form of commodity exchange, via monetary means; or C-M-C, that is, commodities (C) are sold to buy other commodities (C), with money (M) functioning as a means of circulation), rather than understanding and confronting capitalism as a monetary market economy (monetary expansion via commodities, or M-C-M').

A prime example of how the neoclassical C-M-C conception of the economic system became part and parcel of ecological economics is via the adoption of the circular flow model of exchange value. The fundamental idea and worldview of ecological economics is to *extend* the circular flow by adding

raw materials, energy, and biophysical cycles that support the economic cycle. The problem however, is not that the circular flow model does not acknowledge nature, which could then simply be 'added on', but that the circular flow model is flawed in itself. What is needed is a realistic theorisation of value relations that identify capitalism as a spiralling, not circular, system of exchange (e.g., [56]). These foundations can be found explicitly in Marxian Political Economy.

Thus far, the Marxian understanding of value and capital has not entered the core of ecological economics prominently at all. On the contrary, ecological economics has developed as a discipline in which key scholars, who have developed whole research streams within the field, including past presidents of the society, consider the Marxian approach as severely limited for the advance of knowledge in our field (for detailed examples see [45]). As a result, relations between ecological economists and Marxists have been frosty. Ecological economics textbooks are marked by an almost [57–59] or complete absence [60,61] of references to Marxist thinking—the recent Handbook of Ecological Economics is an exception [62]. Marxian thinking remains associated with failed political projects, and whilst this negative reception is partly understandable, it is a fatal mistake to abandon realistic and insightful economic theory for these reasons. On the other hand, there are also many writings about ecological sustainability from a Marxian perspective, which is sometimes termed 'eco-socialism' [63] or 'eco-Marxism'. Early seminal contributions by James O'Connor [64,65] and more contemporary leading examples, such as Andreas Malm's *Fossil Capital* [66] or Jason Moore's *Capitalism in the Web of Life* [67] and essential eco-feminist contributions, such as Ariel Salleh's 'Ecofeminism as Politics' [68] gain importance, as global ecological crisis intensify. However, these contributions remain rather at the fringes in ecological economics, especially in the journal *Ecological Economics* (A Title-Abstract-Keyword search in Ecological Economics for "circuit of capital" yields 0 results; for "capital and Marx" one result [69]; for "capital and Political Economy" 10 results). A few eco-Marxist contributions exist that apply the Marxian understanding of capital (e.g., [70]). One contribution from Paul Burkett [69] stands out in addressing the issue prominently. Burkett aims to defend Marx's reproduction schemes against widespread misconceptions by leading ecological economists (Georgescu-Roegen, Daly, and Martinez-Alier in particular), who portray the Marxist explanation of capitalist reproduction as mechanistic and based on a closed systems approach in which the contribution of nature is taken for granted. It is fair to say that Burkett's classical Marxian understanding of capital and capitalist reproduction, which we share, has not entered the core of ecological economics).

Clive Spash has forcefully promoted the adoption of social theory and political economy to develop ecological economics. In his seminal contribution, *New foundations for ecological economics*, he critiques orthodox economics for failing to address reality and suggests seeing:

> *"The future of ecological economics firmly amongst heterodox economic schools of thought and in ideological opposition to those supporting the existing institutional structures perpetuating a false reality of the world's social, environmental and economic systems and their operation."* [52] p. 36

However, in what follows, Spash fails to establish these new foundations explicitly. He does not mention profits, capitalist competition, surplus, financialisation, or money—at all. 'The economy' is mentioned only once, in the typical ecological economics representation of the economic process—as critiqued above—*"the economy is embedded in the Natural environment and subject to the Laws of Thermodynamics"* [52] p. 43.

Whilst almost all ecological economists critique neoclassical economics from a sustainability perspective, we still lack a clear articulation of what the heterodox alternative is, spelled out from its basics. Instead, many ecological economists keep reproducing mainstream economic concepts and thinking, often unconsciously (as demonstrated above, for example). Letting go of neoclassical economic foundations is more easily said than done. This influences what is studied and, more importantly, what is not. Joan Martinez-Alier and Roldan Muradian for example, propagate in the introductory chapter of one of the latest handbooks of ecological economics that:

*"The study of the market (the chrematistics) should come after the study of ecology and social institutions . . . the market economy could not exist without social institutions, and without the unpaid services of ecosystems."* [71] p. 2

Whilst it is clear that humans cannot exist without functioning ecosystems, this framing is problematic because it guides research away rather than towards a realistic study of—and full confrontation with—capitalism, as a major driver behind global ecological challenges, and as the moving force of the accelerating Anthropocene. The core contradiction of capitalism is exactly that the system operates *as if* it were disconnected from its surrounding ecosystems, resulting in a clash between biophysical and economic realities. By neglecting *"the study of the market"*, as the authors propose, we miss precisely the part that is crucial to understand and change. Important questions and issues are left unanswered, because they often remain unasked. As a result, ecological economists widely neglect deeper seated social drivers of environmental destruction—much in the same way as these drivers are neglected in neoclassical economics.

## 3. Research Priorities Moving Forward

Our vision for ecological economics is to further develop as a critical and radical social science, grounded in an understanding of capitalist dynamics. We advocate a political economy approach that combines systems thinking, realism, holism, interdisciplinarity, and nuance in the study of specific provisioning systems and how they can be improved.

### 3.1. Social Justice, Well-Being, and the Struggle to Achieve Them

In focusing on its analysis of economic interactions with the environment, ecological economics often loses sight of the bigger picture—a bigger picture that development economists understand better. As Amartya Sen has pointed out repeatedly, economics should always be understood as a means to an end—that end is human flourishing [72].

Paradoxically, ecological economists remember that human well-being is important in its own right and different from economic growth, when criticizing economic growth and the centrality of GDP as an indicator of progress [73,74]. But in everyday practice, in our research on environment–economy linkages, we often forget to bring well-being into the picture.

Some researchers stand out in opposition to this trend and have been doing great work in reminding us of different well-being approaches [2,75]. They have highlighted different theoretical approaches, pointed out that well-being is fundamentally multi-dimensional, and that elements of cultural differences become very important in considering how well-being is satisfied [76–79].

The problem here is that we cannot just tack on well-being indicators alongside or instead of GDP in our analysis and be done. As soon as we bring considerations of well-being and human needs and rights into the picture (and we should!), the picture changes, and our research approaches must change as well, in the following ways.

1.　Not all consumption is created equal, as neoclassical utility maximisation posits. This is true from a well-being perspective (clean water for drinking vs. clean water in a private swimming pool, energy used for refrigerating vaccines vs. refrigerating coca-cola, . . . ), and it links closely to other core areas of interest in ecological economics, such as inequality and fair distributions. However, we too often fail to differentiate types of consumption of the same category of product or service within our analysis and policy prescriptions. Taking well-being and human needs seriously means interrogating the purpose and outcomes of consumption [80].

2.　Well-being is social, not individual. When we aspire to create an economy that is a means to the end of achieving well-being, this is a collective statement, not an individual one (unlike, for example, through consumerism). Providing a decent life for each other is fundamentally a collective effort. Clearly, diverse individuals require specific accommodation for their life circumstances and stages (small children, disabled people, women of childbearing age, elderly

people, and so on). However, the provision of well-being is overwhelmingly collective, through health systems, education, culture, communication networks, urban planning, affordable access to life's necessities and so on. This means ecological economists must study production beyond pure material objects and supply chains: we must study collective provision and its enabling (or disabling) of well-being.

3.  Well-being is political, not technocratic. If we take well-being on board, and of course we should, we must also take on board the understanding that achieving universal well-being, or improvements in well-being in any community, is a project involving political struggle. This is a struggle between those who do not have power or access to resources and need more; and those who currently hold power, control resources, and/or have played a historical role in the current unequal status of various communities. A technocratic approach here is disastrous, because it omits power relations. Simply doling out necessities, like food, healthcare, and access to work and education, to communities will certainly result in tangible benefits to those communities. However, durable, ongoing progress can only be made when disenfranchised communities are given back control over the provision of their own well-being. Only an overtly political analysis can raise this issue to the level of importance it requires. Manfred Max-Neef, a founding and towering figure of ecological economics, understood this very well when he designed 'Human Scale Development' action research processes [81–83]. Recently, Hilary Cottam has been pioneering social design processes based on capabilities, in the context of providing welfare differently [84]. Ecological economists can learn more from these approaches and embed our science with more effective policy and politics.

### 3.2. From Mainstream Economics to Political Economy

We suggest learning more from political economists, such as members of IIPPE (the International Initiative for the Promotion of Political Economy), many of whom study how capitalist provisioning of basic services is often biased towards the rich and against the vulnerable, and what alternatives might be envisaged. Such studies require grounding in 'sound' economics. We disagree with colleagues who present themselves as pragmatic about the use of economic theory and methodology. In ecological economics, 'abstract' theory is often contrasted with 'concrete' action and real-world problems, and the latter is what should be prioritised. This is a false dichotomy, because ideas shape how we think about and act in the world. Ideas matter whether they are empirically right or wrong, and economic ideas matter especially because they grant access to resources and power. Power relations, in turn, influence and dictate what ideas gain currency. They shape knowledge production in crucial ways, by channelling what is studied and what is not, as well as what is financially and institutionally supported and what is not. Engaging with theory and methodology is therefore not an abstract exercise in academic ivory towers, but a powerful political instrument, one that cannot be escaped. Being indifferent or pragmatic about the use of theory means buying into mainstream concepts, tools, and narratives, more often than not. But *because* environmental crises are accelerating, we cannot afford to rely on floppy theories.

Theories are not solutions per se and are not everything it takes to change the world. That much is clear. Yet good theories reveal spaces for action, warn of obstacles, and crucially, help us to see beyond empirical reality [85]. In contrast, theories are highly problematic when they hide what is important and prevent sensible actions. There are substantive reasons why heterodox economists speak of the mainstream as Freakonomics, Zombie-economics, Post-autistic economics, or the Economics of the 1%. What unites different heterodox economic traditions is their commitment to debunk neoclassical economics as a serious barrier for the advance of knowledge and a dangerous guide for action. We would like to see ecological economics belong firmly to this pluralist heterodox community. We can be pluralistic, but reject theoretical eclecticism at the same time [86,87]. To be clear—rejecting neoclassical theory does not mean rejecting everything that is proposed by neoclassical (environmental) economists, such as eco-taxes, green investments, or subsidies.

What does it take to promote heterodox development within ecological economics? We see two different avenues. One way is through continued critique. As neoclassical economics remains alive and well and extremely influential within ecological economics (see Section 2.3), grounds remain to contest it. It is not just another school of thought in a pluralist toolkit, but a hegemonic discipline that protects vested interests and core capitalist institutions, rather than critically exposing and dismantling them in the broader societal interest. For example, the consequence of 10 years of austerity politics justified by austerity economics in the city we are based, Leeds, is that 60% of children live in poverty today. Austerity, around the world, is a politically aggressive agenda promoting social destruction, cloaked in technocratic and neutral-sounding neoclassical economic terms. On such grounds, mainstream economics remains a social disaster and core barrier to systemic changes, but one that can be de-mystified and confronted. We see critique as an essential ingredient of transition and expression of active resistance. It helps to create transparency by informing people about unacceptable circumstances, flawed justifications for unequal distribution of resources, and can be combined with building alternatives. It calls researchers to look for what is not spoken and written about, who is left out, and why. This includes debunking economic myths and challenging the dominant narrative where it is false or morally wrong.

Another possibly more powerful way to address the failings of the mainstream is to make it redundant. Better alternatives don't have to be invented, they exist. Heterodox economics, as a broad tent, is the science of the social provisioning process [86] (rather than the science of the allocation of scarce resources under severely constrained conditions, as in mainstream economics). Focusing on social provisioning means understanding how societies organise the flow of goods and services necessary to maintain and reproduce themselves, in the context of historically-specific systems and structures. The historically specific dominant system of provisioning today is capitalism. Heterodoxy is therefore about *"analyzing, theorising, and transforming capitalism . . . to envision transformative policies for the majority, not an elite"* [87] p. 3. The aim is to explain the multiplicity of real economic activities and factors that shape production and consumption processes, and connect the individual to those—such as the structure and use of resources; changes in social wants; production and reproduction of the business enterprise, family, state, and other relevant institutions; distribution; and issues of race, gender, ideologies, and myths.

We advocate Marxian Political Economy as a school of thought for ecological economics. Many Marxian insights are almost universally accepted as core contributions to economics and remain unchallenged to this day. However, the implications of Marxian economics for ecological economics have never been fully drawn out. In *Ecological Economics*, there are surprisingly few contributions that explicitly apply Marxian theory to specific social ecological problems (e.g., [88–92]). There are more contributions that implicitly draw on Marxian thought, such as research grounded in political ecology or eco-feminism. However, most of these applied works do not detail their theoretical underpinnings and, as a result, centrepieces of Marxian Political Economy—an understanding of value and capital, and how they drive and shape the societies we live in—remain under-considered and under-appreciated in ecological economics. This is necessary to understand the intertwined root causes of our predicament, and hence to propose alternative programmes which address these root causes. Many ecological economists shy away from adopting a Marxian perspective—we argue this aversion is most often based in fundamental misunderstandings of Marx's work [45]. Marx had many roles and contributions—for a brilliant and engaging overview of these, for and against, we refer our readers to Heilbroner's *Marx* [93]. Our focus here is Marx's specific goal to highlight the societal consequences of the capitalist mode of production. We could say the main problem with capitalist production is that capitalist societies are too successful. In the first instance, capitalism is a highly productive system. Producing value essentially points to pressures to increase labour productivity (via pressures to lower abstract socially necessary labour time). The resulting highly productive, and seemingly collaborative, global division of labour leads to higher energy and resource use and more work, instead of more leisure time and resource sufficiency, because it is underpinned by the drive to extract surplus value. There are two

conditions for the expansion of capital via the production of surplus value: The expansion of labour capacity (as source of expanding profit), and the expansion of the quantity–flow of money (money creation by banks) [94]. Environmental problems emerge as undesirable by-products of these core dynamics. They manoeuvre societies into increasingly destructive modes of living, and the systemic challenge becomes nothing less than confronting capital as a dominant social force of the system as a whole. Capital is a positive re-enforcing feedback loop that inevitably leads to planetary overshoot, if nothing is done to break it. From a sustainability perspective, it is a problem-generating structure [8]. This understanding is tightly intertwined with an understanding of power and vested interests, and the role of key capitalist institutions such as markets, credit provision, finance, or the State, which illuminates implementation barriers and limits of policy-making.

How can ecological economists take this analysis on board? The task is to follow flows of value throughout the whole system. The Marxian approach provides the framework for understanding how value streams that appear in biophysical, monetary, and other social forms travel together along the whole life cycle—spanning production, consumption, and waste streams. Thus, biophysical analysis can be combined with Marxian value theory. Use value and exchange value flows can be traced, for example, to understand waste or emission flows throughout production and consumption systems (e.g., [95]). Other considerations, including ownership patterns, profit relations between actors, institutional structures, and so on should also be included, for instance through Ben Fine's Systems of Provision approach [35,96]. The Systems of Provision approach guides the study of the structures, agents, relations, and processes that shape different provisioning systems (e.g., the provision of food, water, electricity, housing, or transport). It considers the full chain of activities that underpin the material and cultural (re)production of goods, thereby studying physical and social processes and outcomes in tandem, rather than separating them out. Such analysis can be applied to specific sustainability problems at local, regional, or global levels, by tracing the specific cultural, monetary, historical, and other context-specific circumstances. It thereby helps explain, for example, why some modes of provisioning are good at translating energy and resources inputs into desirable social outcomes and others are not [97]. The Systems of Provision approach exemplifies what it takes to conduct a good political economy analysis of a specific problem [98,99]. Another example comes from attempts to combine biophysical accounting with political ecology analysis, prominently by the Barcelona school of political ecological economics (e.g., [100–102]). More such analyses are needed in ecological economics because this is what it takes to put into practice what has long been claimed necessary for ecological economics: To develop the field further as radical political economy of nature.

### 3.3. Confronting Capital

When ecological economists speak about the need for social ecological transformation and systemic change, what does this mean? What are the main problems? Money? Modernity? The market? Globalisation? Neoclassical economics? For us, grappling with the dominant mode of societal organisation is the necessary realistic starting point. We need nothing less than a civilizational shift, and whether you like it or not, most of the world's civilisation today is capitalist. Change both within and beyond capitalism requires us to comprehend the system as a whole and how it unfolds in specific circumstances. Is this possible, given that reality is complex, contradictory, and constantly changing? Yes, it is. Whilst some aspects of capitalism have evolved in the last 200 years, its core logic is still the same—which is what we need to confront, if we are serious about advocating transformational changes.

In terms of building alternatives, Marxian value analysis brings clarity in identifying urgently needed interventions because it helps to lift the veil of distorted realities we are presented with every day in media, politics, and mainstream economics. Working with realistic theory that does not shy away from pointing to uncomfortable truths helps to create transparency. Demonstrating that the logic of capital—which is, in the simplest terms, production for profit, not need—gives rise to macro dynamics of ecological overshoot, which are diametrically opposed to steady-state or circular economy ambitions advocated in ecological economics, and have practical implications for research,

outreach, and action. Identifying capital as the core problem implies to fully confront capitalism, with a recognition of the need for social struggles at its core.

This is not a reduction of everything to life vs. capital, as the Marxian position is often mischaracterised. Capitalism has benefitted many, including ourselves. It is not a life vs. death dualistic story but a life and death (increasingly death) dialectical story. If we consider the system as a whole, capitalist tendencies such as overproduction, overconsumption, concentration and centralisation, commodification, alienation, and financialisation explain the realities that underpin multiple crises [45]. Acting in accordance with the 'good of the whole' therefore means to transcend capital, rather than feed it. This is not a depolarising perspective but a realistic assessment of the status quo. We can help people see how capital shapes most of what we are doing and see how it dehumanises and delivers somewhat successfully for some. Overall, it is an entrenched system of harm, coupling domination over the environment with domination over people [103,104]. Creating transparency in this way is empowering. We believe that 'seeing clearly' has huge potential to mobilise people to join the shaping of a new paradigm that is being born before our eyes.

What does this imply for building alternatives more specifically? It implies to clarify our role as researchers. We are not the (only or main) ones who build concrete alternatives on the ground, and hence not the ones who offer blueprints for change or lists of universal policies and actions, which is paternalising and ineffective—we may be more like architects and facilitators, who can help formulate design principles of systemic change, translate them to specific contexts, warn from end-of-pipe 'solutions', and create transparency along the way, not least by learning from past (and often non-white) struggles [105]. Examples of frameworks in ecological economics that provide guidance for systemic change include Donella Meadow's Leverage Points [43] or Capra and Jacobsen's 'Principles of Life' [106]. Kate Raworth draws attention to the whole economy of household, communities, industry, and government in her articulation of *Doughnut Economics*, including promoting alternatives that are redistributive (socially) and regenerative (environmentally) by design [41]. A recent promising political example of a systemic change framework includes the US Green New Deal [107,108]. Moreover, alternative social and decision-making approaches, including Max-Neef's Human Scale Development [81,82], Hilary Cottam's social design [84], and even ideas from Bookchin's radical municipalism [103] can and should be brought into our research and activism. Such holistic visions and purpose-driven approaches are eminently suited to being adopted and taken forward by ecological economists.

We can confront capital by prioritising what ultimately matters: satisfying human needs in a context of global justice (see Section 3.1). We know quite well what basic human needs are and how they can be fulfilled (meaningful participation in society, access to health care, healthy food, decent housing, etc.) [77,109,110]. It is essential to steer research and build alternative institutions in directions that focus on delivering such ultimate goals directly, rather than indirectly, via intermediate means such as growth, profit-making, employment- or money-creation. These latter are not ultimate goals, they are means to achieving something else. Focusing on intermediate means often creates side-effects, such as the extremely wasteful use of resources and energy under capitalism, that can be detrimental to the ultimate goal of the satisfaction of needs.

This implies the implementation of alternative models of provisioning essentials—housing, food, transport, health care, education—that are not structurally inclined towards expansion. This could take forms of co-operative production and distribution which emphasise use and access, rather than ownership [111] or the nationalisation or municipalisation of essential industries and bringing them under real democratic control [112]. Provisioning and protecting life should be taken out of the market. Governance principles that allow effective delivery of services for alternative production can be found in energy democracy, citizen's assemblies, or sociocracy.

The core problem is not that alternatives are missing, but lack of support and power to implement them (e.g., media and politics co-financed, disciplined, and lobbied by capital). Confronting capital therefore also implies confronting the capitalist state. The State has played a central role in supporting

and stabilizing capitalism from Day 1. It provides the legal architecture for the reproduction of the system and ensures more or less redistribution of its surpluses, above all via the protection of private property rights in nature, means of production, the employment of labour, and appropriation of surplus value produced by that labour [94,113]. Ecological economists often advocate 'stronger states' to implement sensible policies, without problematizing the role of the state itself. The issue is not big vs. small or strong vs. weak government, but whom government serves.

> *"The idea that conservatives trust the market while progressives want the government is a myth. Conservatives simply are not honest about the ways in which they want the government to intervene to distribute income upward."* [114] p. 14

Governments serve capital by paying for basic infrastructure, essential training and schooling, supporting investment with taxes/subsidies, opening new markets through privatization and trade, and managing labour relations or rescuing businesses in times of crisis [113]. The capitalist state needs to be challenged and criticised much more overtly than is currently done in ecological economics, and powerful state–capital alliances discredited and dismantled if they continue to support environmental and social destruction. The role of social movements and independent research is crucial in this respect, or a combination of both [115].

Academic research can and should no longer be purely academic. Facing ecological and humanitarian catastrophes on a planetary scale requires action and engagement, for instance, by supporting civil disobedience [116–118]. We strongly believe that ecological economics research should be embedded in local, national, and global initiatives and activism, not just abstract analysis or remote guidance. A long tradition of scholars engaged in activism (e.g., [119–122]) needs to be boosted. Academic activism also means that it is not enough for us sit back from the fray and comment that the world should be on a different course, it is now clearly part of our work to actively shift this course, by the means provided to us through our scholarship. We can no longer 'just' be scientists—science is called upon to advocate and act in alignment with its observations and conclusions.

This new, braver and bolder, but also more relevant and promising, type of research will require new types of engagement, collaboration (rather than competition), engaged, active and open teaching, and so on. Editing and reviewing guidance for publications will have to be revisited to insist upon and reward such ambition and consistency. Scientists will look to each other on becoming effective agents (not just observers!) of change: rigour, openness, criticism, peer review all must play their role for robust and rapid progress.

## 4. Conclusions: Roots, Riots, and Radical Change

We advocate an ecological economics that remains the same yet changes substantially. What should remain is the vision and ambition that another world is possible in which all beings live together well within limits. Where we see the need for changes is to consequently translate this ambition into every-day research and action. It means, for instance, shifting our attention from biophysical drivers (which naturalise the problem), and from problematizing growth (which accentuate outcomes), towards confronting the social and political root causes of unsustainability and injustice. It also means letting go of neoclassical economics, which is, upon closer scrutiny, a school of thought designed to protect the status quo by supporting existing industries and elites and delaying meaningful action. We must not be naïve about the possibilities of systemic change. At the same time, we must not be blind or ignorant about what we are up against: Capital, the dominant societal form of organising life around the globe.

> *"So how do you change paradigms? . . . In a nutshell, you keep pointing at the anomalies and failures in the old paradigm, you keep speaking louder and with assurance from the new one, you insert people with the new paradigm in places of public visibility and power. You don't waste time with reactionaries; rather you work with active change agents and with the vast middle ground of people who are open-minded."* [43] p. 18

Aligned with Donella Meadows' vision, we see the role of ecological economists as supporting social struggles on two fronts: Fostering the conditions of the birth of a better future; and fighting against what needs to die so that this future can exist—damaging industries, technologies, and political regimes. Research alone is clearly not enough towards this end. It needs to be an effective communication and outreach tool for radical urgent action. Radical change will not come into being without active resistance, protests, and solidary movements that rise up against unacceptable modes of living and politics. As ecological economists we need to speak up, point to uncomfortable realities, proactively engage with interest groups, and steer decisions in our realms. This is the road less travelled we advocate for ecological economics, because it may make all the difference.

**Author Contributions:** Writing—original draft preparation, E.P. and J.K.S.

**Funding:** We thank the Leverhulme Trust for funding the Living Well Within Limits project.

**Acknowledgments:** Thank you to two anonymous referees for valuable comments.

**Conflicts of Interest:** The authors declare no conflict of interest.

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
