# Peer review of "Roots, Riots, and Radical Change—A Road Less Travelled for Ecological Economics"

_sustainability, doi:10.3390/su11072001_

Round 1
Reviewer 1 Report
The article focusses on topics that are interesting and relevant. There is a need for a discussion that pay attention to the political and systemic basis for ecological economics.
Some comments to the authors;
1. The research question should be formulated more clearly, and the conclusion should be
more closely linked to the research question.
2. The relationship between neoclassical (capitalist) economy and ecological economics should
be argued more clearly. Is neoclassical (capitalist) economy a prerequisite for ecological economics or is
ecological economics an alternative to neoclassical (capitalist) economy?
3. Under point 3.2. Marxist economy is advocated as a political platform for ecological economics. It
could be interesting to get an explanation of the extent to which the challenges outlined in the articleare influenced by the political context in which the ecological economics is introduced.
4. Line 55-57, the concepts must be defined and the connection between them are too vague. Line 66-67. Why is it harmful?
5. William Rees draw a demarcation line between green econo0my and ecological economics, his contribution is relevant.
6. "Anarchism and ecological economics - A transformative approach to a sustainable future" Routledge 2019 (O. Jakobsen) focus on topics relevant for the research question presented in the article.
Author Response
Dear Reviewer,
thank you very much for taking your time to provide comments to improve our article.
Please find attached our responses and the new consolidated manuscript (including changes requested from a second reviewer).
Very best wishes

Reviewer 2 Report
In this paper, the authors outline three major critiques of ecological economics and propose how they can be remedied. The paper, while uneven in parts, makes good and valid points which are often well argued. The critiques regarding overemphasis on growth, the need for more political engagement and the suggestion to adopt a Marxist political economy approach, while not entirely novel, are well presented and much needed. I for one am sympathetic to their perspective and convinced.
My main critique is that due consideration is not given to literature from ecological economics and allied fields that are already adopting the approaches being advocated for. For example, the authors propose for ecological economics to adopt a Marxist political economy approach, however they do not engage at all with the already rich literature on Marxist environmental sociology, a la John Bellamy Foster, Jason Moore, etc. or Marxist eco-feminists such as Ariel Salleh. Further, much of what they propose regarding reconciling political ecology / economy and ecological economics is already a key focus of the Barcelona School of Political Ecological Economics, however references to this vast body of work are extremely thin in the text. Finally many ecological economists are engaged in thinking around scholarship and activism and can be productively referenced there.
In general, the citations fail to stir excitement. For authors calling out for new approaches, they lack diversity. While they include all the key reference names and the giants of the field, they neglect (perhaps strategically to better make their point?) a trove of exciting work happening by scholars in ecological economics and allied fields that can be meaningfully mobilized as showing the path forward. There is the need to gesture more clearly to what the research agenda they are outlining would look like and this would be one way the paper can demonstrate more effectively and clearly how it would address its own admonitions to the field through concrete methodologies and paths forward rather than the current somewhat vague proclamations.
Here some examples (off the top of my head but I would advise to consider this approach throughout):
- The one example offered twice of a valuable approach is Ben Fine´s systems of provision approach - why? What is it and how does it offer a better frame of analysis than others, please draw this out. Are there no other examples of other worthwhile approaches?
- There have been numerous attempts to combine biophysical accounting with political ecology analysis that can either be referenced, or critiqued for their shortcomings as the case may be, but should be addressed. A few here:
Temper, L., 2016. Who gets the HANPP (Human Appropriation of Net Primary Production)? Biomass distribution and the bio-economy in the Tana Delta, Kenya. Journal of Political Ecology, 23(1), pp.410-433.
Muradian, R., Walter, M. and Martinez-Alier, J., 2012. Hegemonic transitions and global shifts in social metabolism: Implications for resource-rich countries. Introduction to the special section. Global environmental change, 22(3), pp.559-567.
Martinez-Alier, J., Temper, L. and Demaria, F., 2016. Social metabolism and environmental conflicts in India. In Nature, Economy and Society (pp. 19-49). Springer, New Delhi.
Schaffartzik, A., Mayer, A., Eisenmenger, N. and Krausmann, F., 2016. Global patterns of metal extractivism, 1950–2010: Providing the bones for the industrial society's skeleton. Ecological Economics, 122, pp.101-110.
- Similarly are the authors the first to critique eco-eco´s growth obsession and how to move beyond it?
- A related critique could be about the boundaries of ecological economics and this relates to the methodology. What falls under eco-eco and what does not? Are they considering the literature based on the journal? As a keyword, etc.? On what basis do they come to these conclusions?
Finally, the section on capitalism and Marx is not entirely convincing. It seems insufficient to me to say that capitalism is the problem and leave it at that. This is another form of depolarization that aims to reduce everything to life vs. capital. I disagree that a Marxist analysis “brings clarity in identifying urgently needed interventions” and would suggest the authors to go deeper beyond promising proposals and alternative models of provisioning such as “co-operative production and distribution which emphasize use and access, rather than ownership [73], or universal basic services [74].” These are complex issues and a research agenda on how to contest capitalism and create alternatives may well be beyond the scope of this paper, but the gravity of the task at hand should not be addressed so flippantly.
Author Response
Dear Reviewer,
thank you very much for taking your time to provide comments to improve our article.
Please find attached our responses and the new consolidated manuscript (including changes requested from another reviewer).
Very best wishes

Round 2
Reviewer 1 Report
I find the revision satisfying and look forward to see the printed version og the article.
Reviewer 2 Report
Thank you I am satisfied with the revisions made and believe the article is currently much stronger.